# Metagenomes from Coastal Sediments of Kuwait: Insights into the Microbiome, Metabolic Functions and Resistome

**DOI:** 10.3390/microorganisms11020531

**Published:** 2023-02-20

**Authors:** Nazima Habibi, Saif Uddin, Hanan Al-Sarawi, Ahmed Aldhameer, Anisha Shajan, Farhana Zakir, Nasreem Abdul Razzack, Faiz Alam

**Affiliations:** 1Environment and Life Science Research Centre, Kuwait Institute for Scientific Research, Safat 13109, Kuwait; 2Environment Public Authority, Fourth Ring Road, Shuwaikh Industrial 70050, Kuwait

**Keywords:** shotgun sequencing, environmental DNA, bacteria, functional annotation, biomonitoring, archaea

## Abstract

Coastal sediments in the proximity of wastewater and emergency outfalls are often sinks of pharmaceutical compounds and other organic and inorganic contaminants that are likely to affect the microbial community. The metabolites of these contaminants affect microbial diversity and their metabolic processes, resulting in undesirable effects on ecosystem functioning, thus necessitating the need to understand their composition and functions. In the present investigation, we studied the metagenomes of 12 coastal surface sediments through whole genome shot-gun sequencing. Taxonomic binning of the genes predicted about 86% as bacteria, 1% as archaea, >0.001% as viruses and Eukaryota, and 12% as other communities. The dominant bacterial, archaeal, and fungal genera were *Woeseia, Nitrosopumilus*, and *Rhizophagus*, respectively. The most prevalent viral families were Myoviridae and Siphoviridae, and the T4 virus was the most dominant bacteriophage. The unigenes further aligned to 26 clusters of orthologous genes (COGs) and five carbohydrate-active enzymes (CAZy) classes. Glycoside hydrolases (GH) and glycoside transferase (GT) were the highest-recorded CAzymes. The Kyoto Encyclopedia of Genes and Genomes (KEGG) level 3 functions were subjugated by purine metabolism > ABC transporters > oxidative phosphorylation > two-component system > pyrimidine metabolism > pyruvate metabolism > quorum sensing > carbon fixation pathways > ribosomes > and glyoxalate and dicarboxylate metabolism. Sequences allying with plasmids, integrons, insertion sequences and antibiotic-resistance genes were also observed. Both the taxonomies and functional abundances exhibited variation in relative abundances, with limited spatial variability (ANOVA *p* > 0.05; ANOSIM-0.05, *p* > 0.05). This study underlines the dominant microbial communities and functional genes in the marine sediments of Kuwait as a baseline for future biomonitoring programs.

## 1. Introduction

The effluents from wastewater treatment plants (WWTPs), power and desalination (P&D) outfalls, and land-based pollutants from other sources have driven significant changes in marine biodiversity [1]. Most WWTPs are not very efficient in capturing pharmaceuticals in wastewater, resulting in the release of metabolites and xenobiotics into coastal waters. The released pharmaceuticals negatively affect the diversity and functions of vital microbial communities [2]. These pharmaceuticals in the aquatic environment affect taxonomic profiles, gene abundances, and metabolic processes [3,4]. Several reports suggest that these coastal sediments receiving pharmaceuticals discharge become enriched in pathogenic microbes [2,5,6,7], posing a significant risk to marine organisms and the human population [8,9].

The correlation between environmental microbes, trace metals, and persistent organic pollutants has been established in a few studies [10,11,12,13]. Most of these studies used microbes and their genetic markers as indicators for environmental pollution, thus lacking information on whole community profiles. Advances in high-throughput sequencing have revolutionized the detection of genes in complex environmental communities, offering a more promising avenue for comprehensive genetic profiling. The taxonomic and functional profiles of microbes have been reported from estuaries, rivers, lakes, coral reefs, mangroves, sediments in coastal areas, and deep marine sediments using a high-throughput sequencing approach [4,12,14,15,16,17,18,19]. Recently, more research has been focused on the environmental DNA whole genome sequencing of sediments receiving pollutants, such as metals, oil discharges, and industrial and domestic waste [12,20,21,22,23,24,25].

This study aims to generate baseline information on microbial diversity in the coastal sediments of the northwestern Persian Gulf, which is a semi-enclosed shallow water body [26] with reasonably high residence time [27,28,29]. Although most of the wastewater in Kuwait is treated, the mean number of fecal and non-fecal coliforms far exceeds the Brazilian legislation standards and thresholds of European coastal bathing water directives (cBWD) [30,31]. Recent investigations have reported pharmaceuticals the treated waste streams near and the emergency outfalls in the coastal waters of Kuwait [32,33]. Antimicrobial resistance genes have also been recorded in these environments [34,35,36]. Bottom sediments act as sinks for these bioactive compounds and impose discerning pressure on the aquatic biota. Mapping the microbiome is, therefore, imperative to assess the threats and impacts associated with historical and contemporary pollutants entering the marine streams of the Persian Gulf [37,38].

Hence, the characterization of microbes in marine/freshwater sediments is of high scientific interest. This study presents the metagenomic profile of surface sediments near emergency outfalls along Kuwait’s coastline. The sequence data were mined for taxonomic distribution and the predominant metabolic functions carried out by the microbial communities. Antibiotic-resistant genes, plasmids, and integrons were also identified. The spatial variations in the relative abundances of microbial communities and metabolic functions were also studied.

## 2. Methods

### 2.1. Sample Collection and DNA Extraction

A total of 12 surface sediment samples (Figure 1) were collected along Kuwait’s coastline from September to October 2021. Each location’s GPS coordinates were plotted on the Kuwait map using the ArcGIS software v 10.4.1 (Esri, Redlands, CA, USA). At each station, a sediment profile of 10–15 cm (grab sample) was collected and packed in 50 mL sterile centrifuge tubes (Corning^®^ Glendale, AZ, USA), followed by transporting on ice to Kuwait Institute for Scientific Research (KISR) laboratories. The samples were aliquoted and stored frozen at −20 °C until DNA extraction [32]. All the sampled sites were in close vicinity of storm outfalls, except S4 and S12. Both these locations were pristine and unaffected by emergency waste disposal. The total DNA from each sample (0.25 g) was extracted using a PowerSoil^®^ DNA Extraction Kit (QIAGEN, Germantown, MD, USA). DNA was extracted multiple times (*n* = 5) from each site and pooled to reach the desired concentration (1000 ng for sequencing). The quantity and quality of the isolated DNA were evaluated using a Qubit fluorometer (Thermo Fisher Scientific, USA) and agarose gel electrophoresis (Bio-Rad, Darmstadt, Germany), respectively, before library preparation (Appendix A; Appendix A). Bacterial cell counts (total prokaryotic cells) per gram of samples were estimated for each sample through a quantitative polymerase chain reaction (qPCR) [36]. The purified DNA (ca.1 µg) was lyophilized (using FDB-8603 Operon, Gimpo, Republic of Korea) and shipped to a sequencing facility for whole genome metagenomic sequencing.

### 2.2. Metagenomic Sequencing

Metagenomic sequencing was performed at Novogene, AIT Genomics, Singapore Ltd. Prior to sequencing, the dried DNA was resuspended in nuclease free water (Ambion^TM^, Carlsbad, CA, USA) and re-checked for quality and quantity, as mentioned above. The DNA was fragmented by sonication and converted to sequenceable libraries through the NEBNext^®^ Ultra^TM^ DNA (Illumina, San Diego, CA, USA) kit. Fragmented DNA was processed through the steps of A-tailing, index ligation, and the addition of Illumina adapters at both the 5′ and 3′ ends of the DNA segments. The libraries were purified using the AMPure XP beads (Agencourt, Beckman Coulter Genomics) and quantified through qPCR (Life Technologies, Carlsbad, CA, USA) [39]. Post-normalization, the libraries were pooled and loaded at a 10.0 pM concentration on an Illumina NovaSeq 6000 platform (Illumina, San Diego) for 2 × 150 bp paired-end sequencing [36].

The raw reads were trimmed and aligned with Bowtie2 v 2.2.4 to remove host contamination [40]. The reads were subjected to FASTQC for initial quality checks [41]. Clean reads were assembled using MEGAHIT v1.0.4 into scaftigs. Scaftigs (≥500 bp) were used for ORF (Open Reading Frame) prediction by MetaGeneMark v 2.10 [42]. The CD-HIT v 4.5.8 was used to obtain the gene catalogue from the filtered ORFs (>100 nt) [43]. Gene abundance was calculated based on the total number of mapped reads and gene length. Taxonomic annotation was performed through DIAMOND v 0.9.9 by aligning the unigenes to the microNR database version 2018-01-02 of NCBI (blast, −e 1e−5) [44]. The aligned sequences were further treated in MEGAN to filter matches < e value *10 [45]. Krona plots were created using the web version of KRONA tools [46]. Differential heat maps were made by applying the Wilcoxon Sum Rank test on median abundances; *p* < 0.05) [47] in MicrobiomeAnalyst [48].

### 2.3. Functional Annotation

Genes were translated to proteins and aligned against the evolutionary genealogy of genes: Non-supervised Orthologous Groups (eggNOG) version 4.1 [49], Carbohydrate-Active enzymes (CAZy) version 2014.11.25 [50] and Kyoto Encyclopedia of Genes and Genomes (KEGG) [51,52,53] databases. The DIAMOND BLASTX (−e 1e−10, best hits reserved) protocol was used to identify the pathways from the above databases. The identified KEGG Orthology (KO) genes were further annotated into different pathways based on predefined collections in the KEGG database and quantified by reading counts. The unigenes were also BLASTP against the standard Comprehensive Antibiotic Research Database (CARD) database (e value ≤ 1e−5) to filter antibiotic resistance gene orthologues (AROs) [54]. Plasmids, integrons, and insertion sequences were picked through alignment with the integral, ISfinder, and plasmid databases v 2018 (−e 1e−10, BLASTN), respectively.

### 2.4. Statistical Analysis

A principal coordinate analysis (PCoA) and analysis of similarity (ANOSIM) were performed in R (ade4 and vegan package, version 2.15.3) [48,55]. Non-parametric multidimensional scaling (NMDS) was also performed on Bray–Curtis distances between relative abundances of microbial communities [48,55]. Packages, including ggplot2 and gplots in R and matplotlib in Python, were used for visualization purposes. Six alpha diversity parameters (Observed, Chao1, ACE, Shannon, Simpson, and Fisher) were calculated on rarified data [48,56].

## 3. Results

Twelve metagenomes generated an average of 6.44 Gb of raw data per sample with a mean read count of 42,940,656. Adapter removal and quality filtering yielded 6.43 Gb of usable data per sample for downstream processing, with an effective percentage of 99.85% (Appendix A). The assembly of raw reads resulted in scaftigs ranging from 63,498,148 to 283,332,790 bp with an average N50 of 806 bp and N90 of 536 bp, respectively, that were subsequently used for gene prediction (ORFs) (Appendix A). Scaftigs above 500 bp in size were only used for ORF prediction, and ORFs less than 100 nt were removed to predict unigenes (Figure 2a,b).

### 3.1. Taxonomic Profiling

The taxonomic binning of the predicted ORFs revealed overwhelming abundances of bacteria (86%). About 1% of the total ORFs accounted for the distribution of the archaeal communities (Figure 3a). The representation of viruses and Eukaryota was less in all the samples (>0.001%), whereas other communities were 12%. The other communities are most likely to be the higher organisms. Taxonomic classification of the bacterial domain revealed the dominance of Proteobacteria (62.69%) and Bacteriodetes (16.02%) at the phylum level. Gammaproteobacteria (37.3%), Alphaproteobacteria (11.63%), and Deltaproteobacteria (10.55%), were the most abundant classes. Order Chromatiales (20.5%) was divided into Woesiaceae (11.84%) and Chromatiaceae (6.33%), which culminated into genera *Woesia* (11.84%) and *Marinobacter* (1.82%) (Figure 3b). Both these genera exhibited maximum RA and were hence considered the most dominant. Among the archaea, the highly prevalent phyla were Thaumarchaeota (41.72%), followed by Euryarchaeota (29.33%) and Bathyarcheota (18.87%). The majority of the archaea remained unclassified at lower taxonomic levels. Among the classified forms, the top three were *Nitrosopumilus* (8.58%)*, Candidatus Nitrosoarchaeum* (2.98%), and *Cenarchaeum* (2.74%) (Figure 3c). Fungi were the supreme domain of Eukaryota, with the greatest abundances shown by phyla Mucoromycota (28.34%), Chytridiomycota (17.6%), and Basidiomycota (17.44%). The topmost fungal genus was *Rhizophagus* (10.4%) (Figure 3d). The shotgun metagenomic approach was able to capture even the low-abundant viral sequences. These sequences originated from the Myoviridae (14.75%), Podoviridae (9.16%), Siphoviridae (8.95%), Polydnaviridae (2.28%), Mimiviridae (0.82%), and Baculoviridae (0.69%) families. A significant proportion (63.12%) of unclassified viruses were also recorded in these samples. T4 virus (2.51%), Bracovirus (2.28%), Gaiavirus (0.9%), Hokovirus (0.82%), and Alphabaculovirus (0.69%) were the only viral genera recorded in the present samples. Sequencing at a higher depth generating 12 Gb of data per sample is recommended to capture microbial communities with lower abundances. The relative abundances (RA) of all the discovered taxa are provided in Appendix A and Appendix A.

### 3.2. Functional Profiles

To infer the functional potential of the sediment samples, the predicted ORFs were aligned against the eggNOG, CAZy, and KEGG databases. The eggNOG database identified 24 clusters of orthologous genes (COGs) categories. COGs matching amino acid transport and metabolism (E-type), and energy production and conversion (C-type) were very high.

A total of 83,706 genes also matched with the CAZy database (carbohydrate-active enzymes). Ubiquitous classes were AA-Auxiliary activities (1408), CBM-carbohydrate-binding modules (15,745), CE-carbohydrate esterase (4319), GH-glycoside hydrolases (33,070, GT-glycosyl transferases (27,107), and PL-polysaccharide lyases (1427) (Figure 4b).

Within the KEGG database, the genes mapped to five level 1 functions of Cellular Processes (92,418), Environmental Information Processing (111,807), Human Diseases (65,387), Metabolism (92,399), and Organismal Systems (32,395) (Figure 5a). Level 2 functions under Metabolism involved xenobiotics biodegradation (26,468), nucleotide metabolism (64,114), metabolism of terpenoids and polyketides (21,112), metabolism of other amino acids (40,143), metabolism of cofactors and vitamins (81,902), lipid metabolism (38,919), glycan biosynthesis and metabolism (23,362), energy metabolism (109,427), carbohydrate metabolism (137,792), biosynthesis of other secondary metabolism (24,826), and amino acid metabolism (14,283). Genetic Information Processing included functions such as translation (50,274), transcription (6024), replication and repair (39,283), folding, sorting, and degradation (31,527). Under Environmental Information Processing, sub-processes such as signaling molecules and interaction (185), signal transduction (51,566), and membrane transport (60,056) were common. Human Diseases included genes related to the functions of substance dependence (467), neurodegenerative diseases (4107), infectious diseases (viral—2370, parasitic—1646, and bacterial—9598), immune diseases (648), endocrine and metabolic diseases (7792), drug resistance (antineoplastic—6899 and antimicrobial—15,698), cardiovascular diseases (4036), and cancers (specific types—3542 and overview—8684). Under the category Cellular Processes, sub-processes such as transport and catabolism (8218), cellular community (prokaryotes—49,467; eukaryotes—95), cell motility (14,160), and cell growth and death (20,458) were recorded. The Organismal System comprised functions such as the sensory system (33), nervous system (3164), immune system (2260), excretory system (1348), environmental adaptation (2979), endocrine system (12,044), digestive system (1540), development (25), circulatory system (1316), and ageing (7686). The RA of Metabolism was highest, followed by Genetic information Processing > Environmental Information Processing > Cellular Processes > Human Diseases and Organismal Systems. Like CAZy and eggNOG, the RA of the KEGG pathways also varied from S1 to S12 (Figure 5b). The chief level 3 KEGG functions (*n* = 10) in the current investigation were ko00230 (purine metabolism), ko02010 (ABC transporters), ko00190 (oxidative phosphorylation), ko02020 (two-component signal transduction system), ko00240 (pyrimidine metabolism), ko00620 (pyruvate metabolism), ko02024 (quorum sensing), ko00720 (carbon fixation pathways in prokaryotes), ko03010 (ribosome), and ko00630 (glyoxylate and dicarboxylate metabolism) (Figure 5c).

### 3.3. Resistome and Mobilome Profiling

Alignment against the integral, ISfinder, plasmid, and CARD databases returned 256,099, 12,857, 283,165, and 25,014 sequences, respectively. These databases filtered integrons (INT), insertion sequences (ISQ), plasmids (PLS), and antibiotic resistance genes (ARGs), respectively. PLS sequences were maximum, followed by INT, ARGs, and IS. These sequences were annotated into 1782 ISQs, 1567 PLS, 609 ARGs, and 167 INTs. The numbers of all the antibiotic-resistance elements differed spatially. The maximum number of PLS, INTs, and ISQs were recorded at S1 and the minimum at S12. ARGs were highest at S6 (169) and lowest at S11 (57). Further interpretations of the gene type, drug classes, and metabolic action of these resistance elements is important. (Figure 6). Apart from the numbers, more intriguing and interesting is the presence of the mobile genetic elements (MGEs) and ARGs indicating the persistence of a mobilome and resistome in the marine environment of Kuwait. Further, we matched the source of ARGs and bacterial species and recorded around 109 genes to have a similar origin (Appendix A). This suggests the distribution of ARGs in other microbial species and communities. However, the question of whether this is due to horizontal or vertical gene transfer requires our further attention.

### 3.4. Spatial Variations

In the present investigation, the sites S4 and S12 (Group B) were less impacted compared to the others (S1–S3, S5–S11—Group A); therefore, we explored the spatial variations between these groups. Sites S4 and S12 clustered together on a heat map suggesting a similar metagenome prevailing at these locations. The hierarchical clustering at domain level identified the primacy of viruses and archaea at S12 and S4 (Figure 7a). A functional hierarchical map also placed S4 and S12 nearby. A higher pervasiveness of level 2 KEGG pathways was seen at S12 (Figure 7b).

#### 3.4.1. Alpha Diversity Analysis

Alpha diversity was measured to examine the species richness (number of taxonomic groups) and evenness (abundance distribution) at each sampling site. Alpha diversity indices are also suggestive of intra-sample diversity. Data rarefaction was performed before estimating the alpha diversity indices. The observed alpha diversity ranged between 3923 to 4785 among S1–S12. Chao1 and ACE were comparable with observed indices. Shannon diversity was highest at station S2 (4.07) and lowest at station S4 (3.83). A Shannon index above 1 indicated higher species richness and evenness. Simpson was near 0.77 and Fisher ranged between 1351 and 1731 (Table 1). Pairwise comparisons of all the indices returned *p* values > 0.05 (non-significant), indicating the alpha diversity to be evenly distributed at all the sampling locations.

#### 3.4.2. Beta Diversity Analysis

Principal coordinate analyses (PCoA) were used to predict if any community structure existed in these sediments. Although microbial communities were seen as three clusters on the PCoA plot (Figure 8a), the analysis of similarity (ANOSIM) returned an r^2^ of 0.05 (*p* < 0.273). This was suggestive of a weak population structure. The variations were 46.4% at PC1 and 27.5% at PC2. Sampling at a higher frequency is recommended to obtain more meaningful conclusions regarding the population structure of these metagenomes. We also noticed that Station S1 appeared as an outlier. Station 1 is in the close vicinity of a major network of hospitals in Kuwait. This outfall receives emergency hospital waste on a daily basis; therefore, unique metagenomes are expected at this site. Interestingly, stations S4 and S12 were also closer to each other. Their pristine locations justify the nearness of their metagenomes. The rest of the stations were grouped as a single large cluster. Our results were further supported by the Shannon and Simpson diversity indices (Figure 8b). Clustering of KEGG level 1 pathway abundance also yielded similar grouping (Figure 8c,d). An ordination analysis was also performed on the RA of ARGs. Unlike the previous clustering, four groups were seen on the PCoA plot. However, sample S1 stayed aloof and S4 and S12 congregated along with S10 (Figure 8e). Locations S9 and S2 also formed a distinct clan and were closer to sample S1. The variation across the first axis was 21.1%, and the second axis was recorded as 16.9%. In parallel with the PCoA, a dendrogram analysis also distributed the stations into four clusters. The samples in each cluster differed (Cluster I: S9-S2-S10; Cluster II: S8-S5-S3-S7; Cluster III: S6-S1; Cluster IV: S4-S12-S10) (Figure 8f).

## 4. Discussion

Shotgun metagenomic sequencing was employed to study the microbial and functional composition of the coastal sediments of Kuwait receiving emergency waste. Next-generation sequencing has gained immense popularity since 2010 for comprehensively profiling microbial populations in polluted environments [57]. The high throughput sequencing approach employed in the present investigation successfully captured microbes from bacterial, archaeal, fungal, and viral domains [18,58,59]. This is the first study reporting the taxonomic profiles of all four microbial communities in the coastal sediments, suggesting the validity of shotgun sequencing in comprehensive biomonitoring. In addition to the taxonomies, we also present the functional profile of these microorganisms. The findings provide baseline information on the metabolic potential of these anthropogenically affected marine environments in Kuwait.

Bacteria (79–92%) were the prime microbial components, followed by archaea > eukaryote > virae. A considerable proportion was denoted (12%) as other. Bacterial, archaeal, and eukaryotic DNA was reported in oil-exposed coastal sediments of the Baltic Sea, with the majority of eukaryotic DNA belonging to the fungal domain [60]. This was similar to our observation, where 99% of eukaryotes were fungi [61,62] and their abundances were lesser than bacteria and archaea. Bacteria and archaea were reported in polluted sub-surface sediments of Priolo Bay receiving industrial waste [20]. Bacteria and archaea dominate the ocean’s biomass and play a crucial role in the production and degradation of organic compounds [57]. Fungi, although of a lower abundance, are known to play an important role in the recycling of nutrients [60,61]. Viruses are a less explored sediment community. Breitbart et al. [63] reported that viruses are extremely abundant in marine sediment, and most of them are double-stranded DNA phages. Danovaro et al. [64] reported the overall biomass of the top 50 cm of the ocean seafloor (1.74 pg C; 1.5 ± 0.4 × 10^29^) to originate from bacteria (78%; 3.5 ± 0.9 × 10^28^ cells), archaea (21% 1.4 ± 0.4 × 10^28^ cells), and viruses (<1%; 9.8 ± 2.5 × 10^28^). Bacterial counts ranging between 10^3^ to 10^9^ per g of a sample have previously been reported in these sediments [36]. The bathymetric patterns showed that the abundance and biomass of bacteria decreased with depth, whereas viruses and archaea were not affected [64]. This creates further interest in examining the microbes present along the sediment profile of Kuwait Bay, which has numerous outfalls and high sedimentation rates [65,66,67]. Apart from sediments, bacteria and archaea have been found in the surface waters of the South China Sea [68], urban backwaters of Muttukuda (TN), India [1], and deep-sea sediments of Hadal Mariana Trench [69].

Kuwait’s marine area receives a variety of pollutants through local and regional sources. The main discharges are oil-based, sewage-based, desalination activities, ship waste dumping, and dredging, to name a few [70]. The most abundant bacterial phyla observed in the sub-surface marine sediments of Kuwait were Proteobacteria (42–59%). Similar observations were recorded in metal-contaminated sediments of Liuli river [25], and hydrocarbon-polluted coastal marine basin sediments of Priolo Bay [20] and the Northern Adriatic Sea [71]. Microplastic is also a known contaminant in the water bodies of Kuwait [72,73]. This also accounts for the prevalence of species involved in hydrocarbon degradation, such as *Marinobacter* and *Woeseia* [74]. Genera similar to the present study were reported from polluted marine sediments in Italy [20]. Euryarchaeota and Thaumarcheota terminating in the genus *Nitrosopumilus* were also among the most dominant archaeal community in oil-contaminated coastal sediments of the Baltic Sea [60]. Unlike Ascomycota and Basidiomycota reported in sea sediments of the Antarctic Ocean, in the present samples, Mucormycota was the most common fungal phyla [21]. The T4 virus of Myoviridae is a bacteriophage and most probably infects the inherent bacterial communities [75,76].

Due to extensive nutrient input both from land-based sources and atmospheric deposition, the oceanic productivity in the northern Gulf is quite high [74,77,78,79]. A recent study conducted by the Centre of Environment, Fisheries and Aquaculture Sciences (CEFAS) reported an increase in dissolved nutrient concentrations in the past three decades [37]. Effluent discharges through storm outlets introduce a variety of metabolites that act as multiple stressors/nutrients for the inherent microbial community [32,80]. A high abundance of genes (eggNOG) involved in energy production and conversion, as well as amino acid transport, were thus documented in this study. These results were corroborated by our observations on high abundances of GH and GT enzymes (CAZymes), the vital components of cellular metabolism and carbon cycling [81]. Similar results were recorded in the Brazos-Trinity Basin subsurface sediments [82] and Hadal Biosphere at the Yap trench [22]. This suggests the involvement of microbes in heterotrophic processes, such as the degradation of carbohydrates, hydrocarbons, and aromatics. Further, the dominance of proteins involved in signal transduction, mobilome, and defense mechanisms is attributed to the involvement of microbes in swarming motility, antibiotic resistance, virulence, conjugal plasmid transfer, and biofilm formation [83].

KEGG annotations revealed the involvement of microbial species in a complex of level 3 metabolic processes. Effluent discharges in the marine environment are responsible for unique geochemistry [84], empowering the enrichment of certain metabolic pathways. The purine and pyrimidine metabolism pathways can be linked to DNA synthesis [24], as the microbes might be involved in replication and multiplication. The ABC transporters couple with the ATP hydrolysis to actively transport nutrients by sensing environmental changes [85]. The two-component signal transduction system is activated in changing environmental conditions and initiates processes such as enzymatic catalysis, gene expression, and protein–protein interactions [86]. The pyruvate metabolism is particularly active in bacteria in states of excess carbon, where they use pyruvate as a substrate to generate acetate to recycle NAD^+^ and coenzyme A [87]. Aerobic organisms carry out oxidative phosphorylation to oxidize nutrients and release energy in iron-limited conditions [88]. Quorum sensing in marine microbes has been associated with processes linked with the h-carbon cycle and trophic interactions through anthropogenic changes such as ocean acidification and rising sea temperatures [89]. Carbon fixation pathways in marine prokaryotes are also linked to global carbon cycling [90]. Ribosomes play a major role in genetic information processing and protein translation. Carter et al. reported the role of the 30S subunit of ribosome in the decoding and translocation of antibiotics [91]. The glyoxylate and dicarboxylate metabolism essentially maintain the gain or loss of hydrogen ions in a buffered environment. These pathways were also observed in the sediments and surrounding seawaters of the Qinhuangdao mariculture coastal area in North China [24], and in the backwaters of Muttukadu, Tamil Nadu, India, receiving domestic sewage and industrial effluents [1].

In addition to the above, we also detected ARGs at all the sampling locations. We believe their prevalence is due to the selective force imposed by the pharmaceuticals and antibiotics discharged into the ocean through wastewater streams [32,33,36,58]. Analogous findings were reported in polluted marine environments of the South China Sea [2], the Gulf of Kathiawar, and the Arabian Sea [18,19,58]. ARG abundances associated with mangrove sediments were predicted to be higher in Asian countries [19]. The presence of plasmids, insertion sequences, and integrons, along with ARGs, generates further concern due to their roles in disseminating ARGs across the marine ecosystem via horizontal gene transfer (HGT) [92,93,94]. Pathogens recurrently acquire new resistance genes from environmental species [95]. More intriguing is the presence of these genetic elements at clean beaches not receiving significant waste discharges. The preponderance of proteins such as AcrB [96] and InsO [97] involved in muti-drug efflux systems and mobilomes also requires deeper investigation into the resistors and mobiles of these environments, with a focus on their roles in horizontal and vertical gene transfer.

The effluent discharges into marine streams and the geochemical characteristics at each sampling location shape the microbial community structure [84]. Variations in the relative abundances of microbial communities were thus observed. However, the species richness and evenness were at comparable levels between different sites (measured through the alpha diversity analysis). The community structure in the present samples followed a weakly heterogeneous pattern (ANOSIM r^2^ < 0.1; *p* > 0.05). One possibility of the limited genetic diversity among the microbial communities is the small sample size collected from a narrow coastline of 499 km/310 mi. Another determining factor might be the multiple stressors arriving and residing at the offshore sediments through the storm outlets [32,33,98]. As expected, station S1 appeared as an outgroup in the close vicinity of major hospitals in Kuwait. This site previously recorded the highest concentrations of pharmaceuticals and antibiotics [32]. Stations S4 and S12 are grouped together. Both these stations were away from the outfalls and relatively pristine. Conducting a comprehensive survey with more samples collected in different seasons would be prudent.

## 5. Conclusions

Shotgun metagenome analysis is a powerful tool to gain knowledge on the microbial community composition, metabolic potential, and resistance profile of natural environments. The prevailing environmental conditions and effluent discharge define the community composition in the present samples. The spatial variations are attributed to the physicochemical status defined by the contaminant deposition at each site. Further analysis could be focused on implementing pollution metrics and the socio-economic status of the region on the diversity of microbes and their functions. The concept of seasonality should also be incorporated.

## Figures and Tables

**Figure 1 microorganisms-11-00531-f001:**
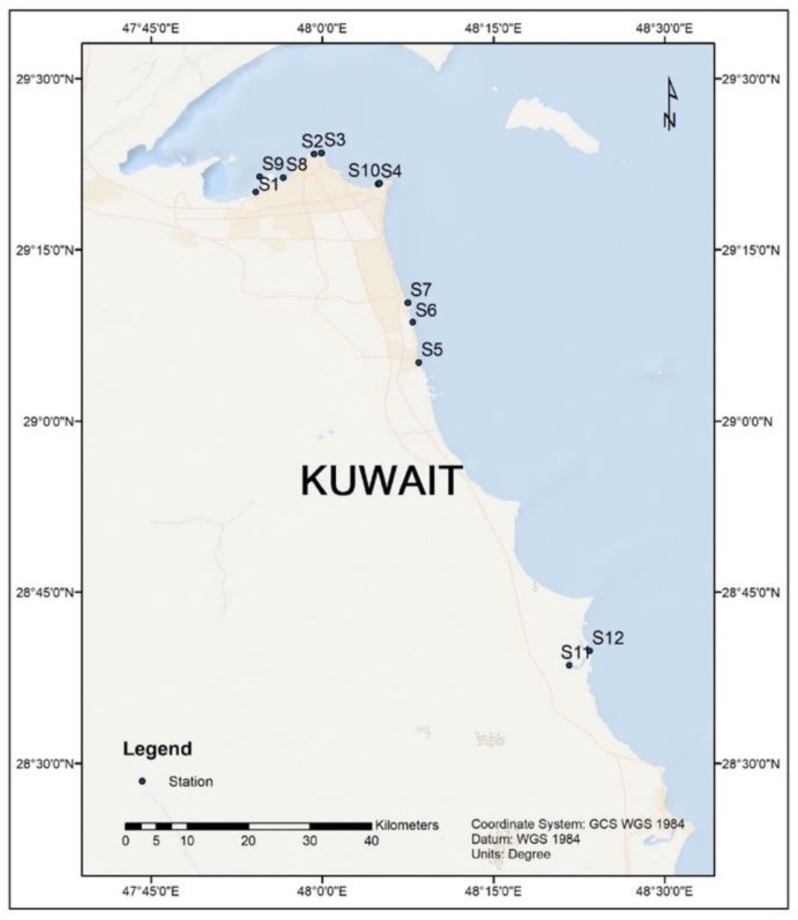
Sampling locations. The GPS coordinates of each location were plotted on Kuwait’s map based using the ArcGIS software v 10.4.1.

**Figure 2 microorganisms-11-00531-f002:**
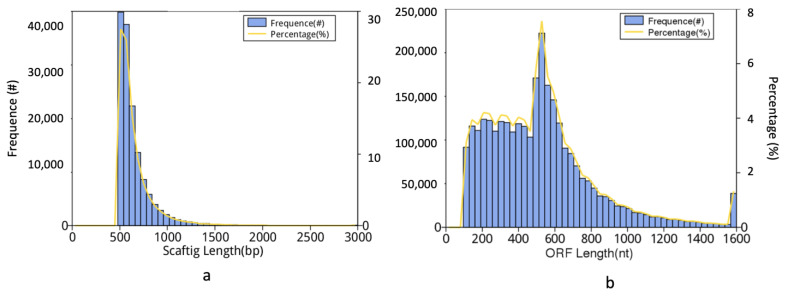
(**a**) Scaftig length distribution; scaftigs > 500 bp were used for open reading frame (ORF) prediction. (**b**) ORF length distribution plot. The ORFs < 100 nt were removed.

**Figure 3 microorganisms-11-00531-f003:**
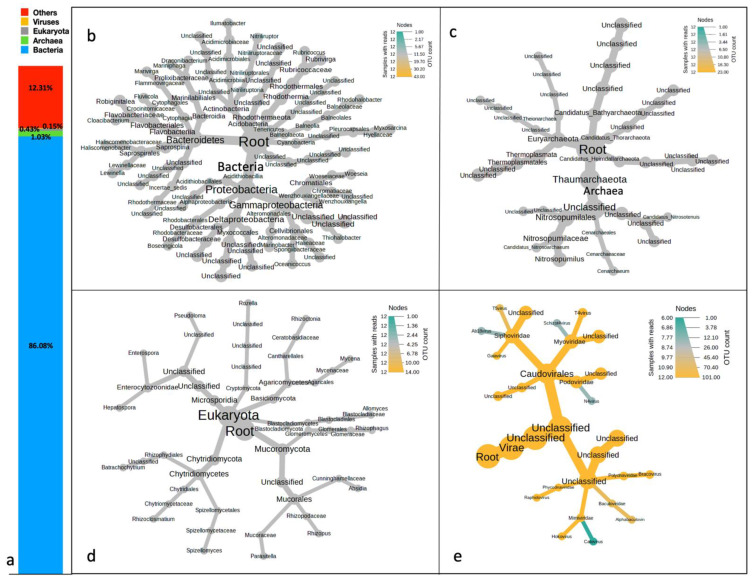
(**a**) Domain-level classification of microbial communities of marine sediments of Kuwait. Taxonomic profiles of (**b**) bacterial, (**c**) archaeal, (**d**) eukaryotic, and (**e**) Viral domains distributed across the marine sediments of Kuwait. Heat trees were created on median abundance by applying Wilcoxson’s sum test at a *p*-value less than 0.05.

**Figure 4 microorganisms-11-00531-f004:**
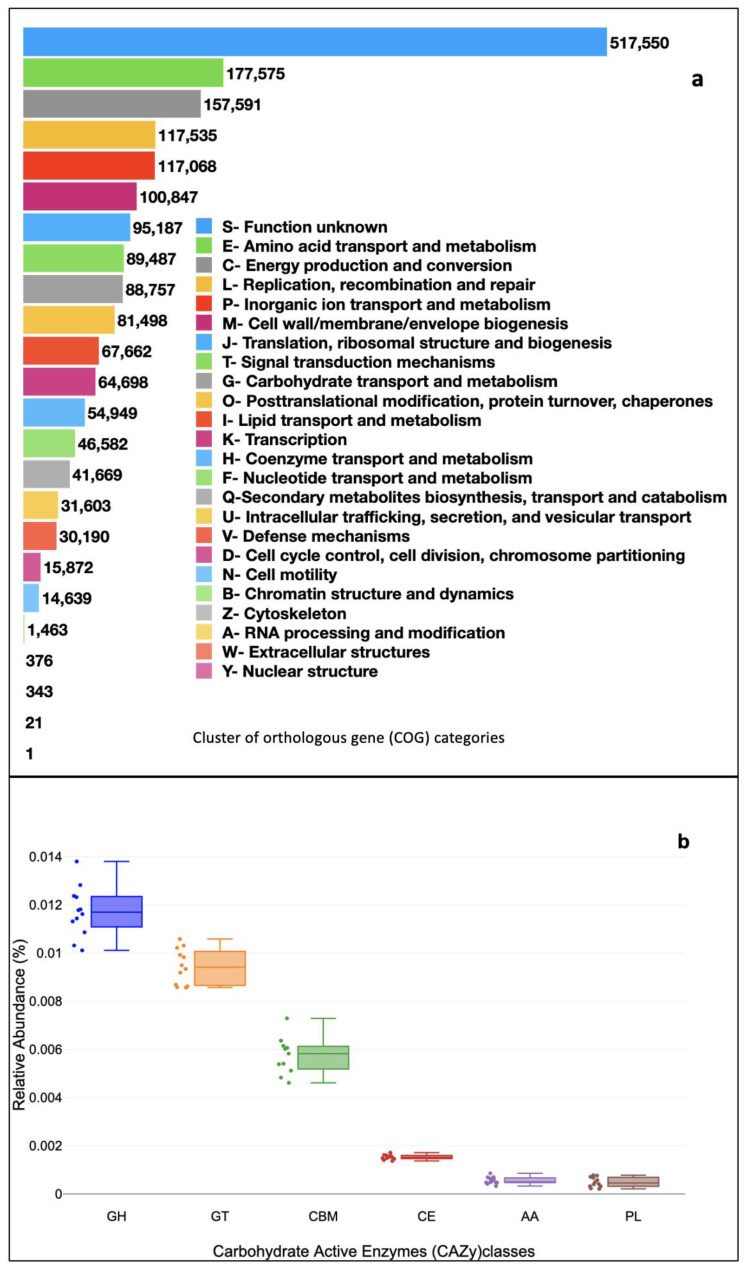
(**a**) Abundance of cluster of orthologous gene (COG) categories filtered through eggNOG database. (**b**) Level 1 carbohydrate-active enzyme classes filtered through (CAZy) database. *y*-axis shows the relative abundances (%), and the corresponding gene categories are plotted on the *x*-axis. For each gene category, a box and whisker plot is drawn. Boxes represent the interquartile range (25–75%), upper and lower whiskers −10–90%; dashed black lines are median abundances. Dots are the abundance.

**Figure 5 microorganisms-11-00531-f005:**
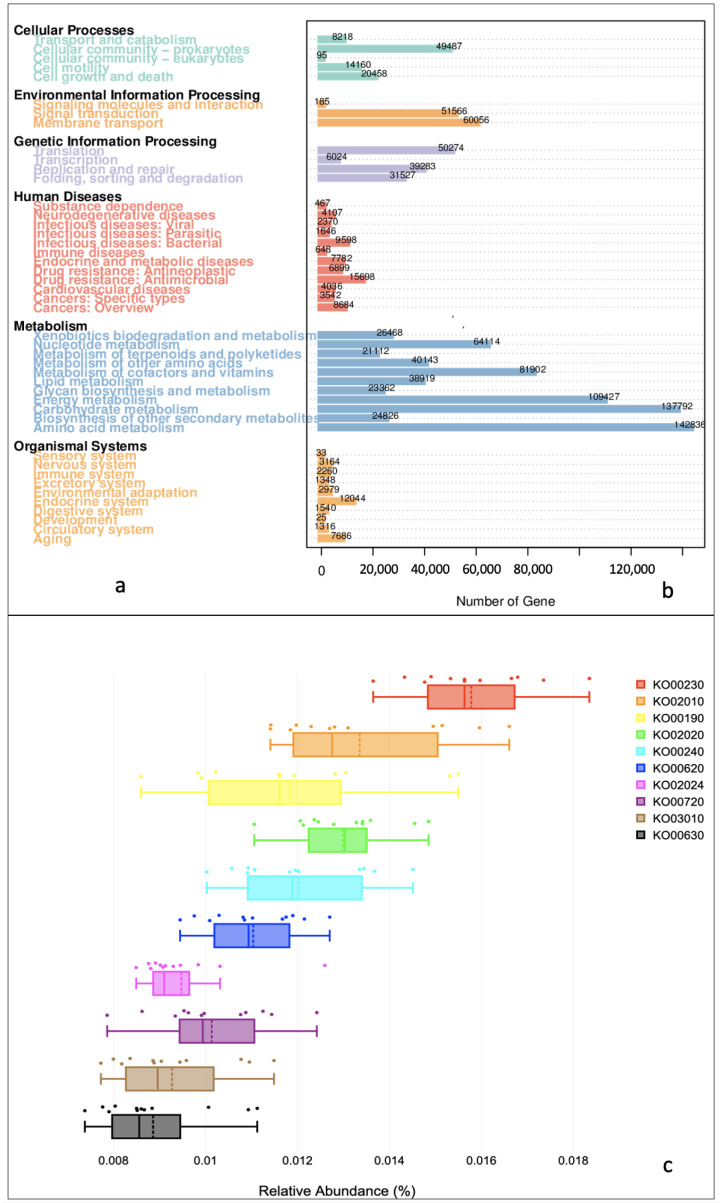
Functional annotation by KEGG database. (**a**) Level 1 and 2 metabolic pathways detected in marine sediments of Kuwait. (**b**) Number of genes associated with level 2 KEGG pathways. (**c**) Box plots showing relative abundances of top 10 level 3 KEGG pathways.

**Figure 6 microorganisms-11-00531-f006:**
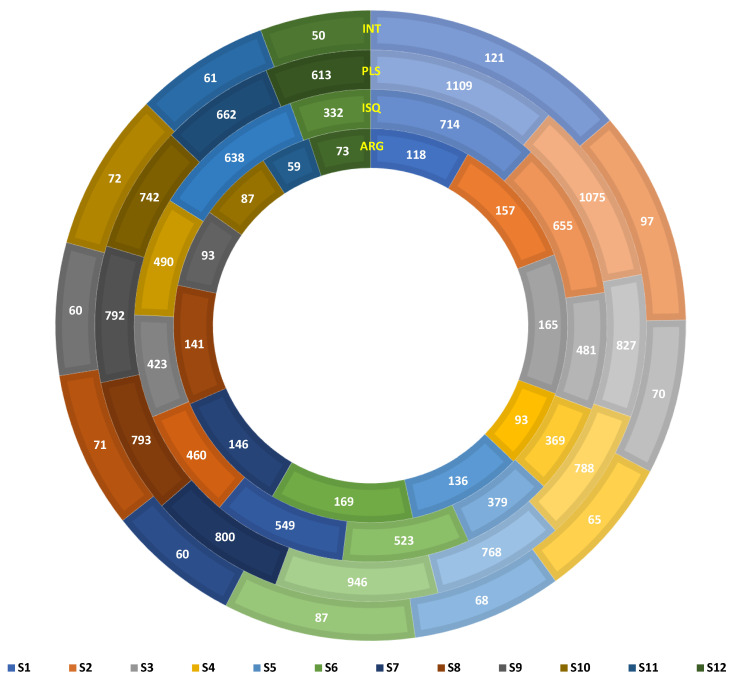
Antibiotic resistance gene elements detected in surface sediments of Kuwait. The circles from outermost to innermost represent the integrons (INT), plasmids (PLS), insertion sequences (ISQ), and antibiotic-resistant genes (ARGs), respectively. The sampling locations are color-coded. A color index is given at the bottom.

**Figure 7 microorganisms-11-00531-f007:**
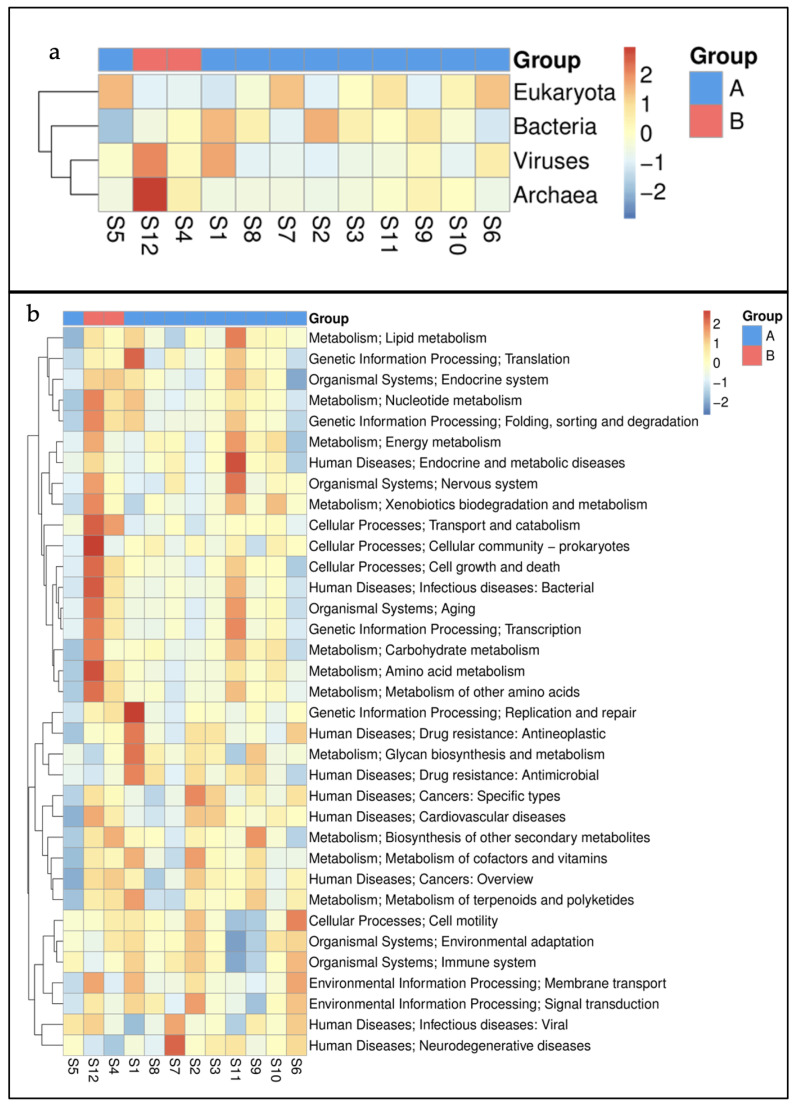
Hierarchical clustering of (**a**) major domains and (**b**) level 2 KEGG pathways in marine sediments of Kuwait. Group A represents the sites near the outfalls and group B are relatively pristine locations.

**Figure 8 microorganisms-11-00531-f008:**
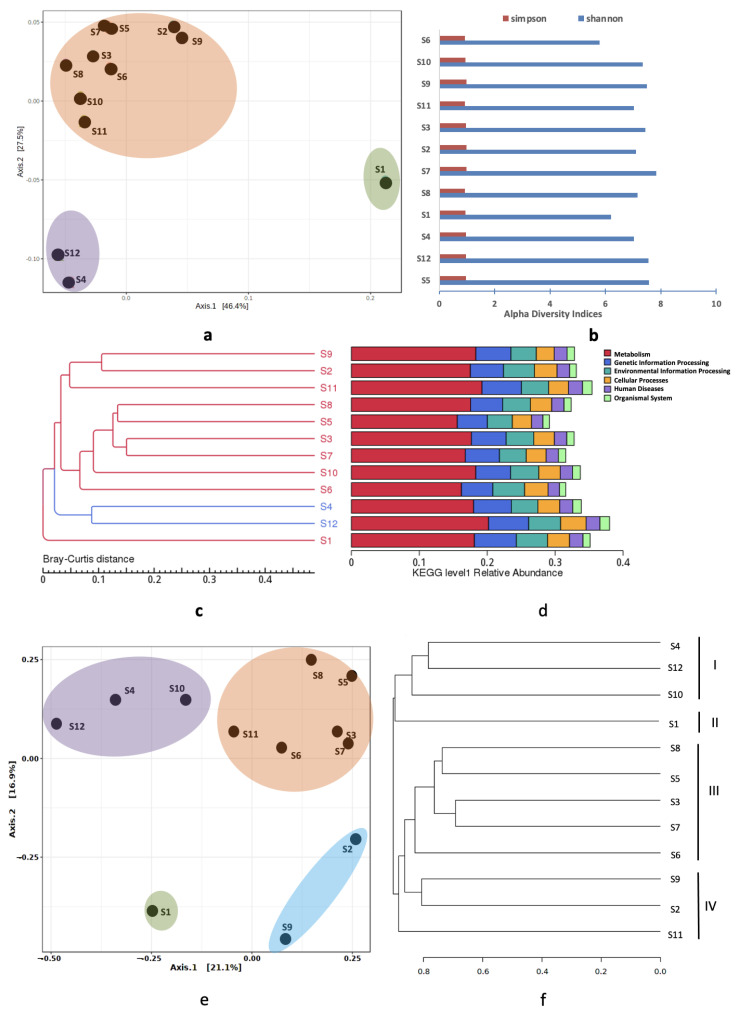
**Spatial variations at taxonomic and functional levels** (**a**) Ordination analysis on RA of microbial taxa. The colors represent the three clusters (**b**) Alpha diversity indices of Shannon and Simpson. (**c**) Dendrogram and (**d**) Cluster of KEGG level 1 RA. Stations S4 and S12 are shown as blue lines. (**e**) Ordination analysis based on RA of ARGs. The four clusters are shown in different colors (**f**) Dendrogram based on RA of ARGs filtered from the marine sediments of Kuwait.

**Table 1 microorganisms-11-00531-t001:** Alpha diversity indices of microbial diversity in marine sediment samples.

Sample ID	Observed	Chao 1	ACE	Shannon	Simpson	Fisher
S1	3923	4269	4272	3.94	0.776	1352
S2	4645	4953	4888	4.07	0.782	1647
S3	4774	5051	5036	4.05	0.787	1609
S4	4457	4707	4711	3.83	0.772	1470
S5	4677	4907	4887	3.85	0.764	1598
S6	4785	5065	5026	3.84	0.752	1731
S7	4721	5006	4990	3.90	0.768	1576
S8	4763	5038	4979	4.01	0.796	1590
S9	4478	4828	4717	4.11	0.789	1433
S10	4613	4880	4857	3.94	0.781	1493
S11	4394	4645	4590	3.96	0.788	1351
S12	4452	4696	4666	3.86	0.772	1440

Observed, Chao 1, and ACE- account for the species richness; Chao 1 and ACE also take into consideration counts of unobserved species; Shannon, Simpson, and Fisher account for both richness and evenness.

## Data Availability

The raw sequences of these data are deposited in the public repository of the National Centre for Biotechnology Information under the accession number PRJNA819259 (SRR18461109-SRR1846120) (https://www.ncbi.nlm.nih.gov/bioproject/PRJNA819259, accessed on 1 January 2023).

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
