# Peer review of "Metagenomes from Coastal Sediments of Kuwait: Insights into the Microbiome, Metabolic Functions and Resistome"

_microorganisms, 2023, doi:10.3390/microorganisms11020531_

Round 1
Reviewer 1 Report
The work is mainly at a description-only level, with poor discussion and not clear interpretation/conclusions, as also evident by the abstract. English and scientific level is poor. The doi link to the supplementary file does not work. Thequality of the figures il low (too difficult to read). The Authors should do a much greater effort to make this dataset feasible for publication. However, the same sequencing dataset has been already published by the same authors (https://www.ncbi.nlm.nih.gov/bioproject/PRJNA819259 and https://www.mdpi.com/2071-1050/14/13/8029 ). So, it should be made clearer that this dataset has been already published, and explain the different work done here. Overall, I think major revisions are needed before considering this work for deeper peer review.
Author Response
Reviewer 1 |
|
Comments |
Response |
The work is mainly at a description-only level, with poor discussion and not clear interpretation/conclusions, as also evident by the abstract. |
We are quite sorry if the reviewer felt that the manuscript is unclear and poor in discussion. We have tried revising it. The manuscript describes the metagenomes of the surface sediments in close proximity to emergency outfalls in Kuwait. The rationale behind emanates from our previous work that showed presence of pharmaceuticals and antibiotics in the marine environments, presence of antibiotic resistance genes at these locations as well. Shot gun metagenomic sequencing was therefore, performed on 12 sediment samples.In the present manuscript, we have provided an insight into the microbial diversity in terms if identifiable bacteria, fungi, archaea and viruses and their metabolic potential. Linkage and interpretations were made with the plasmid and integron databases. |
English and scientific level is poor. |
We are quite perplexed with the comment on the linguistic quality the reviewer has made. The revised version was shared with three native speakers Dr. Will le Quesne, Dr. David Verner Jefferey and Ms. Nicola Coyle from CEFAS, UK. Hope the reviewer will find it acceptable now. |
The doi link to the supplementary file does not work. |
It has been checked, working with us. |
The quality of the figures is low (too difficult to read). The Authors should do a much greater effort to make this dataset feasible for publication. |
The figures have been modified for clarity. Two figures have been added. |
However, the same sequencing dataset has been already published by the same authors (https://www.ncbi.nlm.nih.gov/bioproject/PRJNA819259 and https://www.mdpi.com/2071-1050/14/13/8029 ). So, it should be made clearer that this dataset has been already published, and explain the different work done here. Overall, I think major revisions are needed before considering this work for deeper peer review. |
We have indicated the data is available at this links. In our previous publication this dataset was mined to identify the ARGs, their mechanism of action and drug classes. In addition, we also reported the occurrence of ESKAPEE pathogens in low abundances at same locations. In the present manuscript, the dataset was used to mine the microbial community composition (bacteria, fungi, archaea and viruses) and their metabolic potential. The sequences were also aligned to plasmid and integron databases. As expected, their presence at these sites suggests their roles in horizontal gene transfer of these genes to non-resistant microbial communities and probably to other marine ecosystems. |

Reviewer 2 Report
The authors describe microbial community structures and potential metabolic process in the coastal sediments of northwestern Persian Gulf, based on whole genome shot-gun sequencing. The methods employed in this study seem reliable, please consider the following suggestions to improve this paper.
1. More relevant references regarding coastal sediments subject to anthropogenic influences should be added to the Introduction section.
2. Based on the grouping results of the beta diversity analysis, whether significant metabolic/ARGs difference also exist among these three groups?
3. Relations between the microbial community composition and environmental parameters such as salinity, depth, pH and effluent discharge need further explored to come to the authors’ final conclusion.
Author Response
Reviewer 2 |
|
Comment |
Response |
The authors describe microbial community structures and potential metabolic process in the coastal sediments of northwestern Persian Gulf, based on whole genome shot-gun sequencing. The methods employed in this study seem reliable, please consider the following suggestions to improve this paper. |
We are thankful for the support and appreciation of the reviewer. We believe the shot gun analyses is a reliable method. We appreciate the suggestions of the reviewer, which have been taken into consideration in the revised version. Specific responses are appended below. |
More relevant references regarding coastal sediments subject to anthropogenic influences should be added to the Introduction section. |
Thanks for the suggestion, references have been updated in the revised manuscript. |
Based on the grouping results of the beta diversity analysis, whether significant metabolic/ARGs difference also exist among these three groups? |
Thanks for the suggestion. In the revised version we have included an additional figure on beta diversity analysis of metabolic ARGs. Additionally, two heatmaps are provided to map the metabolic diversity at these 12 sites. |
Relations between the microbial community composition and environmental parameters such as salinity, depth, pH and effluent discharge need further explored to come to the authors’ final conclusion. |
We very much appreciate the suggestion. However, since we have used the sediments, these parameters are not discussed. However the salinity of Kuwait marine area is between 42.3 and 44.1 ‰, the pH varies between 7.93 – 8.24 and the depth at outfall in low tide conditions is 0.0 m while in high tides it can go upto 2.0 – 3.0 m. |

Round 2
Reviewer 1 Report
The Authors have performed sufficient revision. Please check the title "Resi-S-tome. In Figure 3 (the round dendrograms) the color legend is not clear, why are some nodes/parts of the dendrograms yellow/gray/green?
Figure 4 and others, please increase the font size of the numbers/letters and keep a better balance betwenn the written parts and the graphic components of the figure, to allow the reader to read.
The conclusion (L512) "The limited spatial variations are attributed to the contaminants brought in by the emergency wastes" is still not clear. How can contaminants be responsible for a "limited variation"?
Author Response
The Authors have performed sufficient revision. Please check the title "Resi-S-tome. In Figure 3 (the round dendrograms) the color legend is not clear, why are some nodes/parts of the dendrograms yellow/gray/green?
Response: Thank you for bringing this to our attention. We have corrected the spelling of resistome. A colour key has been added to fig 3.
Figure 4 and others, please increase the font size of the numbers/letters and keep a better balance betwenn the written parts and the graphic components of the figure, to allow the reader to read.
Response: These figures are software generated. We cannot do much about them. However, we have enhanced the image quality so as to make the fonts clearer and readable.
The conclusion (L512) "The limited spatial variations are attributed to the contaminants brought in by the emergency wastes" is still not clear. How can contaminants be responsible for a "limited variation"?
Response: We have rephrased this statement

Reviewer 2 Report
I don't have further comments on the paper and think it is ready for publication.
Author Response
We thank the reviewer for accepting the manuscript.